# Extensive Sampling Provides New Insights into Phylogenetic Relationships between Wild and Domesticated *Zanthoxylum* Species in China

Xue Chen [1,2,†], Lu Tian [1,2,†], Jieyun Tian [1,2], Gang Wang [3], Xia Gong [4], Shijing Feng [1,2,*] and Anzhi Wei [1,2,*]

1   College of Forestry, Northwest A&F University, Xianyang 712100, China; chenxueya@nwafu.edu.cn (X.C.); t1anlu@nwafu.edu.cn (L.T.); tianjieyun@nwafu.edu.cn (J.T.)
2   Research Centre for Engineering and Technology of Zanthoxylum State Forestry Administration, Xianyang 712100, China
3   Guizhou Academy of Forestry, Guiyang 550005, China; tltltl@nwafu.edu.cn
4   Sichuan Academy of Botanical Engineering, Neijiang 641200, China; sjfeng@gzu.edu.cn
*   Correspondence: shijingf@nwafu.edu.cn (S.F.); weianzhi@nwafu.edu.cn (A.W.)
†   These authors contributed equally to this work.

**Abstract:** *Zanthoxylum*, belonging to the Rutaceae family, is widely distributed in tropical and subtropical regions. The genus has high economic value as spices, oils, medicinal plants, and culinary applications. *Zanthoxylum* has a long history of domestication and cultivation in China. However, the phylogenetic relationships and origin of wild and cultivated *Zanthoxylum* species in China remain largely unknown. Moreover, there is still no clear molecular phylogenetic system for *Zanthoxylum* species. Herein, 373 *Zanthoxylum* samples were collected from all presently known provenances of *Zanthoxylum* in China. In this study, four chloroplast DNA (cpDNA) markers (*mat*K, *ndh*H, *psb*B, *rbc*L) were used to comprehensively analyze the genetic diversity, relatedness, and geographical origin of Chinese *Zanthoxylum* species. The results were as follows: (1) The aligned length of the four pairs of cpDNA sequences was 3836 bp, and 68 haplotypes were identified according to 219 variable polymorphic sites, including 90 singleton variable sites, 129 parsimony informative sites, 3 Indels (insertions and deletions). (2) Phylogenetic tree and haplotype network strongly supported the division of *Zanthoxylum* species consistent with the taxonomic recognition of five species: *Z. bungeanum*, *Z. piasezkii*, *Z. piperitum*, *Z. armatum*, and *Z. micranthum*. (3) Divergence time estimation suggested that *Zanthoxylum* genus originated from the Late Eocene, and most *Zanthoxylum* species diverged after the Middle Miocene. (4) Haplotype 16 (H16) was at the bottom of the phylogenetic tree, had higher haplotype diversity (*H*d) and nucleotide polymorphism (*P*i) than other haplotypes, and was located in the center of the network figure. Therefore, we deduced that the cultivated *Zanthoxylum* species may originate in Zhouqu County, Gansu Province, China. Meanwhile, our research provided a scientific basis for the identification and breeding programs of Chinese *Zanthoxylum* species.

**Keywords:** *Zanthoxylum*; cpDNA; genetic diversity; phylogenetic relationships; geographical origin

## 1. Introduction

*Zanthoxylum* genus belongs to the Rutaceae family and consists of 250 species worldwide, including 45 species and 13 varieties in China [1]. *Zanthoxylum* is generally dioecious, but only a very few are monoecious and have apomixis characteristics [2]. The genus contains a variety of chemicals such as alkaloids, amides, flavonoids, lignans, terpenes, etc. Therefore, it is a promising source of natural insecticides [3] [4]. Therefore, it has attracted wide attention and research from domestic and foreign scholars. In addition, *Zanthoxylum* also contains calcium, phosphorus, iron, and other trace elements; these trace elements can form a variety of enzymes, play a very important role in human life

activities, can promote the body's metabolism, and enhance the body's immunity (Xiong et al. 2018). Furthermore, the various parts of different *Zanthoxylum* species are used for making medicine to prevent toothache and treat colds, stomachache, and snake bites [5]. Meanwhile, *Zanthoxylum* has been used to develop new antitumor drugs [6] Guo et al. 2010). Most *Zanthoxylum* species have been used as spices in cooking due to the special aroma and flavor of pericarps, mainly owing to the tingling oral sensation created by alkylamines [7]. According to the China Condiment Industry Association, demand for processed *Zanthoxylum* products might increase by more than 20% each year, with annual demand exceeding 100,000 tons [8]. In addition, the *Zanthoxylum* species are suitable for planting on the mountain. It has gradually become one of the most important economic tree species in the project of converting farmland to forest in China. Hence, *Zanthoxylum* has high economic value and ecological value.

Z. *bungeanum* is native to southwestern China and has been cultivated for more than 2000 years. Wild *Z. bungeanum* was covered in Sichuan, Gansu, Shaanxi, Shanxi, Henan, and the mountain regions of Hunan and Hubei. To date, numerous excellent local cultivars have been cultivated during the process, such as Dahongpao (*Z. bungeanum*). Traditionally, the *Zanthoxylum* species have always been identified by their color, shape, fragrance, and mature period. For example, *Z. bungeanum* and *Z. armatum* belong to different *Zanthoxylum* species, but they are cultivated *Zanthoxylum* (Figure 1). People usually identify which species they belong to according to the color of their pericarps. However, the classification of species of *Zanthoxylum* is difficult when based solely on morphological characteristics. Furthermore, different *Zanthoxylum* species have the same name due to their similar morphological characteristics. Synonyms, homonyms, or incorrect historical records may also affect scientific research and result in financial loss. Throughout the long history of domestication and cultivation, a large number of exceptional local *Zanthoxylum* cultivars have been developed. At the same time, the morphological and agronomic characteristics of these *Zanthoxylum* cultivars also gradually began to change. Feng et al. (2016) untangled the evolutionary relationship between the group of *Zanthoxylum* species and inferred that *Z. bungeanum* and *Z. armatum* probably diverged during the Late Miocene, and the ancestors of *Zanthoxylum* first colonized Yunnan and Guizhou provinces [9]. However, there is no reasonable explanation for the origin of *Zanthoxylum* cultivars in China. Thus, it is necessary to understand the origin of cultivated *Zanthoxylum* species and the phylogenetic relationships of wild and cultivated *Zanthoxylum* species in China using more effective and specific methods, such as chloroplast DNA (cpDNA) markers.

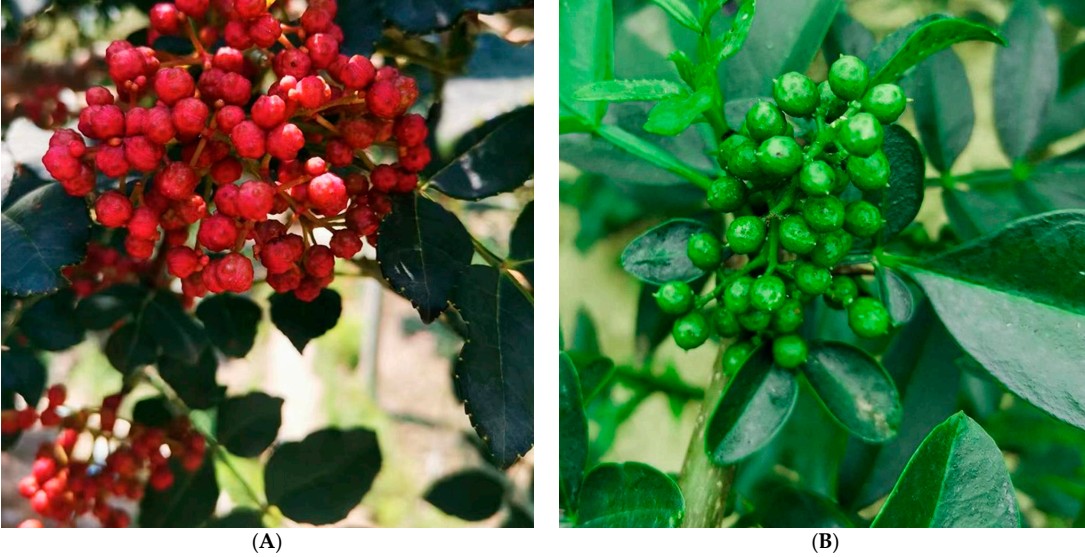

(A)　　　　　　　　　　　　　　　　(B)

**Figure 1.** Photograph of *Zanthoxylum*. (**A**) The fruits of *Z. bungeanum*; (**B**) the fruits of *Z. armatum*.

The genetic properties of cpDNA are generally higher conservation than nuclear DNA and maternal inheritance in angiosperms [10], and there is a relatively high nucleic acid replacement rate in noncoding regions [11], so it is mostly used for species identification, species phylogenetic relationship research and species kinship analysis, as well as in molecular lineage geography and species origin research [12,13]. In recent years, cpDNA markers have been used extensively in systematic studies of other plants for investigating genetic relationships. For example, Wang et al. 2021 analyzed the genetic diversity of sweet potato germplasm based on three cpDNA spacer sequences, *trn*L-*trn*F, *trn*H-*psb*A, and *trn*T-*trn*L [14]. Phylogeny of *Flemingia* was inferred from molecular data of three cpDNA plastid markers (*mat*K, rps16-trnQ, and *psb*A-*trn*H) [15]. The genetic diversity and population structure of the endemic species of Baikal Siberia *Oxytropis triphylla*, *O. bargusinensis*, and *O. interposita* were studied for the first time based on the nucleotide polymorphism of intergenic spacers *psb*A–*trn*H, *trn*L–*trn*F, and *trn*S–*trn*G of chloroplast DNA [16].

Herein, based on a wide-ranging sampling across China, we employed cpDNA markers to analyze the phylogenetic relationships between wild and cultivated *Zanthoxylum* species. The main objectives of this study were to: (i) estimate the genetic diversity of *Zanthoxylum* species and cultivars; (ii) determine the evolutionary relationships among *Zanthoxylum* species; (iii) resolve the geographical origin and spread routes of Chinese *Zanthoxylum* species.

## 2. Materials and Methods

### 2.1. Plant Materials

The fresh leaves of *Zanthoxylum* were collected from all presently known provenances of *Zanthoxylum* in China, including Gansu, Sichuan, Shandong, Shanxi, Shaanxi, Guizhou, Yunnan, and Henan. Silica gel desiccants (HG/T2765.4-2005, Qingdao Haiyang Chemical Co., Ltd., China) were used to dry the fresh leaves of *Zanthoxylum*. A total of 373 *Zanthoxylum* accessions of 103 cultivars were included in this study (Table S1). The collected samples were morphologically recognizable, and the name of each cultivar was designated by local farmers. Spatial coordinates were registered using a Global Positioning System (Table 1). ArcGIS was used to draw the sampling map (Figure 2).

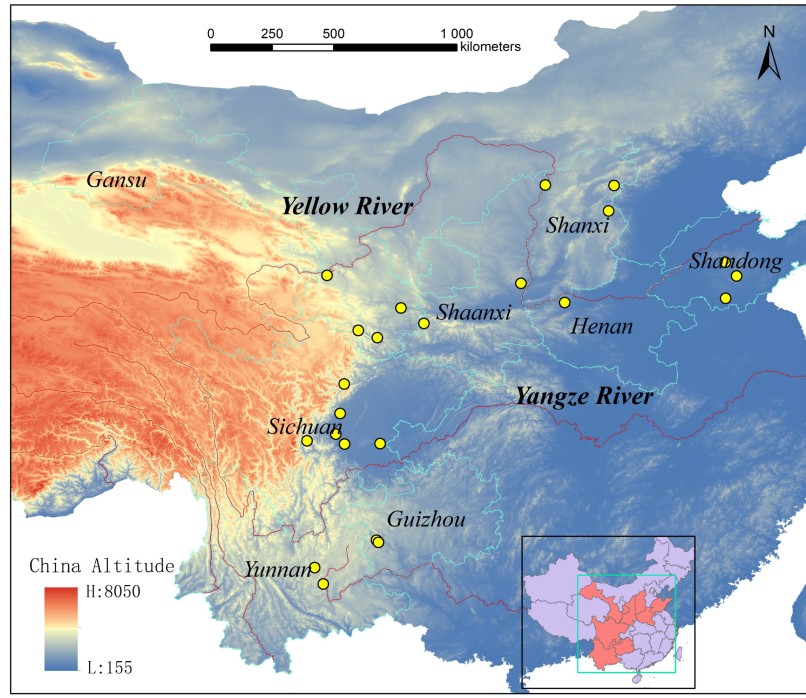

**Figure 2.** Distribution of 373 *Zanthoxylum* samples in 23 sampling sites in China. Small filled yellow circles represent the sampling site.

**Table 1.** Areas of sample collection along with GPS data and altitudinal information.

| Province | Cities | Latitude/N | Longitude/E | Elevation/m |
|---|---|---|---|---|
| Gansu | Tianshui | 34.54 | 105.82 | 1399.38 |
| | Longnan | 33.46 | 104.96 | 1284.08 |
| | Tibetan Autonomous Prefecture of Gannan | 33.72 | 104.26 | 2291.82 |
| | Hui Autonomous Prefecture of Linxia | 35.73 | 103.12 | 1836.13 |
| Sichuan | Ya'an | 29.68 | 102.39 | 2025.83 |
| | Leshan | 29.56 | 103.76 | 365.01 |
| | Neijiang | 29.58 | 105.06 | 335.00 |
| | Aba Tibetan and Qiang Autonomous Prefecture | 31.76 | 103.75 | 2113.09 |
| | Chengdu | 30.68 | 103.60 | 514.91 |
| | Meishan | 29.93 | 103.44 | 464.66 |
| Guizhou | Liupanshui | 26.04 | 104.93 | 1446.93 |
| | Qianxinan Buyi and Miao Autonomous Prefecture | 25.96 | 105.00 | 1477.01 |
| Shaanxi | Weinan | 35.44 | 110.21 | 1027.83 |
| | Baoji | 33.97 | 106.66 | 1243.09 |
| | Yulin | 39.04 | 111.10 | 934.21 |
| Shandong | Linyi | 35.71 | 118.09 | 313.53 |
| | Zaozhuang | 34.89 | 117.69 | 197.16 |
| | Laiwu | 36.21 | 117.68 | 197.17 |
| Shanxi | Xinzhou | 39.02 | 113.61 | 1875.00 |
| | Yangquan | 38.09 | 113.41 | 948.36 |
| Yunnan | Kunming | 25.04 | 102.67 | 2112.25 |
| | Yuxi | 24.44 | 102.98 | 1643.95 |
| Henan | Sanmenxia | 34.74 | 111.81 | 476.00 |

## 2.2. DNA Extraction, PCR Amplification, and Chloroplast DNA Sequencing

The modified cetyltrimethylammonium bromide procedure was used for extracting genomic DNA from 0.070 to 0.075 g of ground leaves [17]. Four pairs of primers were selected for amplification (Table 2). Polymerase chain reaction (PCR) amplifications were carried out in 50 μL reaction mixtures containing 25 μL of 2 × Taq Master Mix (CWBIO Biotechnology Beijing Co., Ltd., China), 1.0 μL of genomic DNA (100 ng/μL), 0.5 μL of each primer (100 μmol/L), and 23 μL of double-distilled water. The conditions for PCR reactions were as follows: initial denaturation at 94 °C for 3 min; followed by 30 cycles of 30 s at 94 °C, 30 s at 56 °C (according to the optimum annealing temperature of primer), and 1.3 min at 72 °C; and then, a final extension step of 7 min at 72 °C, and 4 °C forever. PCR products were run on 0.8% agarose gels (Beijing Laibo Runke Biotechnology Co., Ltd., Beijing, China) by electrophoresis and sequenced by Tsingke Biotechnology Co., Ltd. (Xi'an, China).

**Table 2.** Information of four pairs of cpDNA primers in this study.

| Primer Name | Forward Primer (5′-3′) | Reverse Primer (5′-3′) | Length |
|---|---|---|---|
| *mat*K | MF:CAACACGACTTCCTATACCCAC | MR:CAAATACCAAATCCGCCCTC | 1100 bp |
| *ndh*H | NF:TATCTGCCTTATGTAACCCGTTG | NR:TTTTGGGGCTTCTACTCTCAC | 816 bp |
| *psb*B | PF:TATCGTGTTCATACCGTCGT | PR:CGTGTCCAAAAGTAAACCAG | 1351 bp |
| *rbc*L | RF:CACAAACAGAGACTAAAGCG | RR:CAAAGATCTCGGTCAAAGCA | 1200 bp |

## 2.3. Data Analyses

The sequences of all the DNA fragments with four pairs of primers (*mat*K, *ndh*H, *psb*B, *rbc*L) were edited and manually aligned in Notepad++. DnaSP 6.0 software was used to examine the genetic diversity, haplotype diversity (*H*d), nucleotide diversity (*P*i), polymorphic variant sites, and haplotypes [18]. Subsequently, the pattern of genetic structure and evolutionary relationships among *Zanthoxylum* species were assessed using three different methods. The maximum-likelihood (ML) and neighbor-joining (NJ) methods with 1000 bootstrap were based on the Tamura 3-parameter model in MEGA 7 [19], and also MrBayes for Bayesian inference (BI) phylogenetic tree reconstruction, using *Taddalia*

*asiatica*, *Citrus limon*, and *Citrus sinensis* as outgroups [20]. For MrBayes analysis, four chains, each with a different starting seed, were run for 10 million generations, sampling every 1000 generations. The first 4000 trees were discarded as burn-in. A 50% majority-rule consensus tree was calculated. The phylogenetic tree of haplotype was reconstructed using BEAST. The first 10% of the calculated generations were deleted as burn-in [9].

In addition, we also used DnaSP to carry out Tajima's *D* and Fu's *Fs* tests and used BEAST 1.8 to deduce the evolutionary relationship between 68 haplotypes [21]. In order to initialize BEAST, jModelTest and the Akaike information criterion were used to obtain the appropriate nucleotide substitution models, favoring a GTR + I + G model for each alignment. We tested whether a molecular clock could be fitted to our data by comparing the ML value with and without the molecular clock constraints implemented in MEGA7. The null hypothesis of equal evolutionary rate throughout the tree (strict molecular clock) was strongly rejected (with clock, lnL = −4890.756; without clock, lnL = −3276.201; $p < 0.0001$). Thus, an uncorrelated log-normal relaxed-clock model was chosen. Thus, a Yule process and an uncorrelated log-normal relaxed-clock model were treated as trees prior. We conducted five independent BEAST runs with Markov Chain Monte Carlo (MCMC) simulations of 100 million iterations with sampling every 1000 generations, following a burn-in of the initial 20% cycles. The five runs were then combined in LogCombiner. The ESS of each parameter was greater than 200. The maximum clade credibility tree (MCC) was generated with Tree Annotator 2.4, with the initial 10% of trees discarded as burn-in [22].

Genealogical relationships between cpDNA haplotypes were constructed using NET-WORK v.4.2.0. 1. The program collapses sequences into haplotypes and calculates the frequencies of the haplotypes in the samples. The statistical parsimony was calculated for pairwise differences until the probability was greater than 95%. In this analysis, indels were regarded as one single character resulting from a mutation event. Finally, a neighbor-net tree was built from the concatenated dataset of 68 haplotypes with SplitsTree4. The parameters of split transformation, distance transformation, character transformation, variance, and bootstrap replicates were set to equal angle, neighbor net, uncorrected *p*, ordinary least squares, and 1000, respectively. Other parameters followed default settings [23].

## 3. Results

### 3.1. Sequence Characteristics and Haplotype Detection

The aligned length of the four pairs of cpDNA sequences was 3836 bp, 219 variable polymorphic sites (total number of mutations: 255), including 90 singleton variable sites, 129 parsimony informative sites, 3 Indels (insertions and deletions). Haplotype diversity (*Hd*) was 0.724, and nucleotide diversity (*Pi*) was 0.00203; Tajima's *D* = −2.51347, $p < 0.001$; Tajima's *D* = −2.51347, $p < 0.001$ (Table S2). 68 Haplotypes were recorded as H1–H68, with 24 *Z. piasezkii* (H10–H12, H14–H18, H20–H23, H25–H31, H64), 20 *Z. armatum* (H6, H33–H39, H41, H42, H48–H51, H54, H57, H60–H62, H68), 16 *Z. bungeanum* (H1–H5, H8, H13, H19, H40, H56, H58, H59, H63, H65–H67), 7 *Z. piperitum* (H7, H9, H43–H45, H52, H53), and 1 *Z. micranthum* (H55). Of the 68 haplotypes, nearly half (46/68) were unique (haplotypes that were represented by only one accession), only 9 haplotypes represented by more than 1 accession were found in multiple provinces, and the remaining 13 haplotypes were always confined to a single region. For *Z. armatum*, the haplotype H1 was the most numerous (193/373), including wild and cultivated *Zanthoxylum* accessions from all sampled provinces. H2 and H4 were cultivated *Zanthoxylum*, H2 was found in Gansu, and H4 was found in Sichuan provinces. H6 and H38 were derived from cultivated accessions, while only H54 was found in wild accessions. For *Z. piasezkii*, wild H16 was found in Gansu, and H22 was cultivated *Zanthoxylum* found in Sichuan provinces, but only wild H10 and H14 were found in Gansu. For *Z. piperitum*, H9 was mainly cultivated in Gansu and Sichuan, while others were found in Gansu or Sichuan. Additionally, other haplotypes only occurred in a single province. A great deal of regionally unique haplotypes indicated that most cultivars were probably developed directly from local wild populations, while only the elite cultivars could have the opportunity to be introduced elsewhere.

### 3.2. Genetic Diversity and Phylogenetic Relationships

In this study, *Z. piasezkii* had the highest haplotype and nucleotide diversity ($H$d = 0.992, $P$i = 0.00174), while *Z. bungeanum* had the lowest ($H$d = 0.317, $P$i = 0.00033) (Table S2). However, Gansu, Shanxi, and Yunnan showed the highest haplotype diversity ($H$d = 1.0000) for *Z. armatum*, Henan harbored the highest haplotype diversity ($H$d = 1.0000) for *Z. bungeanum*, and Sichuan harbored the highest haplotype diversity ($H$d = 1.0000) for *Z. piasezkii*, which is probably caused by the high amount of introduced cultivars from other areas. However, the *Z. bungeanum* var. *pubescens* accessions in Sichuan and *Z. piasezkii* accessions in Shandong both had only one single haplotype; thus, their haplotype and nucleotide diversity were both zero. Moreover, *Z. bungeanum* gradually became single during the long-term cultivation process, so the level of genetic diversity of *Z. bungeanum* was reduced. The genetic diversity of some cultivated *Zanthoxylum* species was high. This may be because *Zanthoxylum* had started apomixis.

To confirm the phylogenetic relationship between wild and cultivated *Zanthoxylum* accessions, we constructed neighbor-joining (NJ) and maximum-likelihood (ML) trees [23]. The results were relatively consistent and showed that 373 *Zanthoxylum* samples were divided into five groups with a self-display value greater than 70% in the NJ cluster tree (Figure 3). Group I was *Z. armatum*, referred to as 'Green *Zanthoxylum*', including Qinghuajiao, Zhuyehuajiao, Tengjiao, Dingtanhuajiao, Shuanglingjiao, and Yaojiao. Group II was *Z. micranthum*, including Xiaohuahuajiao. Group III was *Z. piasezkii* introduced from Japan, including Goujiao, Yehuajiao, Chuanshanhuajiao, Yejiao, Yangjiao, and Niujiao. Group IV was *Z. piperitum*, including Putaoshanjiao, Huashanjiao, Jinlingjiao, Zhaocanghuajiao, and Meiguijiao. Group V was *Z. bungeanum,* which is widely cultivated in China, referred to as 'Red *Zanthoxylum*', including Dahongpao, Honghuajiao, Bayuejiao, Huangjinjiao, Meihuajiao, and Dangchanghuajiao.

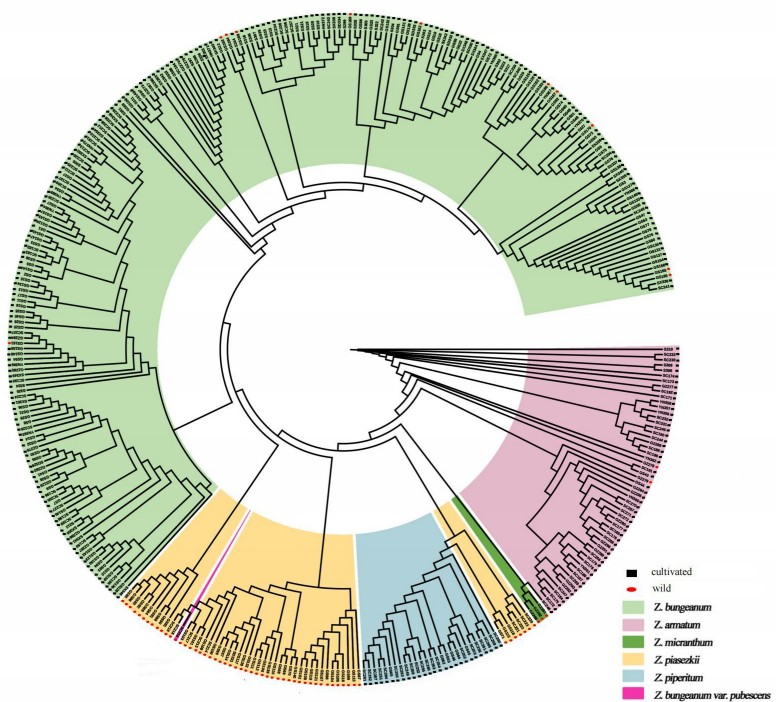

**Figure 3.** Neighbor-joining (NJ) tree of 373 wild and cultivated *Zanthoxylum* accessions based on the cpDNA sequences.

### 3.3. Genetic Structure

To better understand the genetic structure among haplotypes, the MCC tree from BEAST analysis showed that 68 haplotypes were separated into 9 highly divergent and strongly supported clades; this result was roughly similar to the NJ and ML cluster trees

(Figure 4). Clade I *Z. piperitum*, consisted of seven haplotypes derived from Gansu and Sichuan (H44, H53, H52, H9, H45, H7, H43). *Z. piasezkii* was mainly divided into four branches. Clade II included six haplotypes also derived from Gansu and Sichuan (H32, H12, H11, H24, H46, H47). Clade IV included eight haplotypes derived from Gansu, Shandong, and Sichuan (H14, H10, H64, H30, H16, H18, H15, H17). Clade VI included five haplotypes derived from Gansu (H27, H20, H25, H23, H21). Clade VIII included four haplotypes derived from Gansu (H28, H29, H26, H31). *Z. bungeanum* was divided into two parts, while the first part was again divided into three subclades, including ten haplotypes derived from all sampling provinces (H67, H1, H3, H63; H5, H59, H58; H4, H56, H2), and the second portion, including ten haplotypes derived from Gansu, Sichuan, and Shaanxi, namely northwest and southwest China (H40, H65; H8, H19; H13, H66). Haplotypes of *Z. armatum* could be mainly classified into two subclades, including 11 haplotypes derived from Shaanxi, Sichuan, Guizhou, and Gansu, namely northwest and southwest China (H61, H60, H62, H50, H48; H49, H6, H42; H35, H34, H51), and 9 haplotypes were derived from Guizhou, Sichuan, Yunnan, and Shanxi, namely north and southwest China (H38, H37, H41, H33; H39, H36, H57, H68, H54).

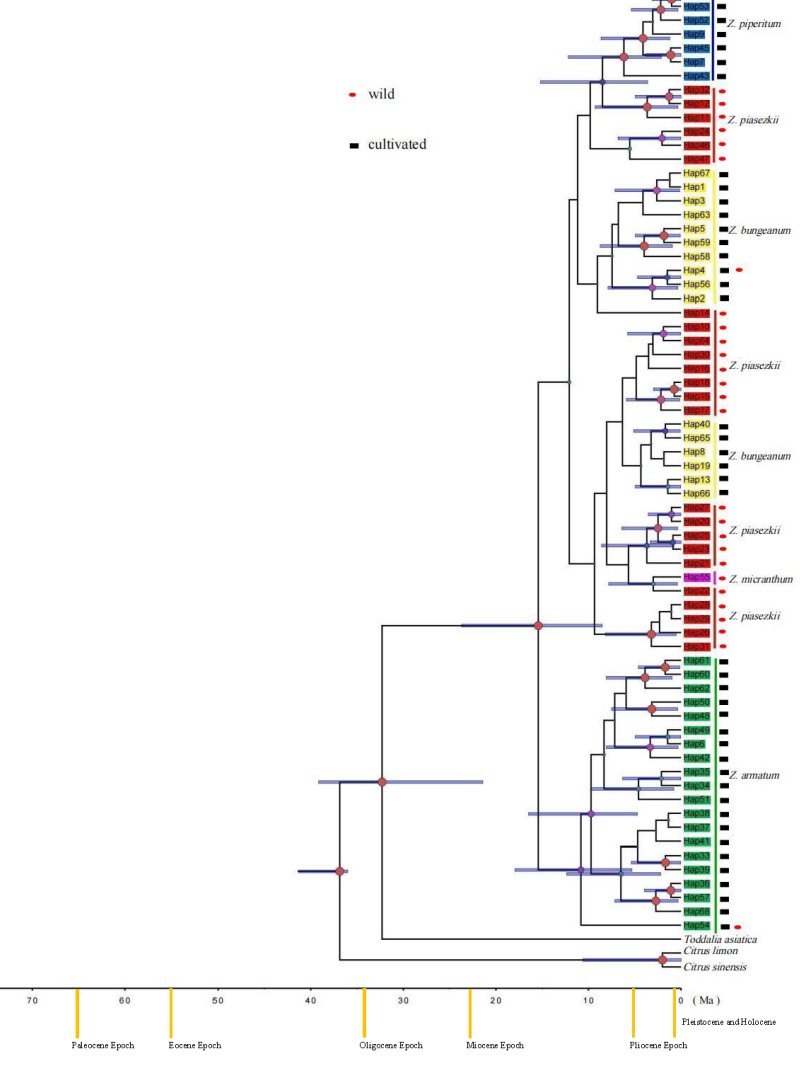

**Figure 4.** The maximum clade credibility tree (MCC) was generated based on the dataset of 68. cpDNA haplotypes of 373 *Zanthoxylum* accessions. *Taddalia asiatica*, *Citrus limon*, and *Citrus sinensis* were used as outgroups. Posterior probabilities based on maximum-likelihood (ML) analysis are labeled above nodes. Blue bars represent the 95% highest posterior density (HPD) credibility intervals for node ages (in Myr ago, Ma).

In addition, we provided a generation time of four years and calculated the divergence time by BEAST [24]. The results showed that the most recent common ancestor of Chinese *Zanthoxylum* plants was about 31.66 Ma, the most recent ancestor of the haplotypes of *Zanthoxylum* collected in this study was approximately 15.82 Ma (8.53–23.72), and the split between *Z. piperitum* and *Z. piasezkii* occurred an average of 8.92 Ma (3.61–15.19). *Z. micranthum* and *Z. piasezkii* occurred at approximately 3.50 Ma (0.40–7.83 Ma) (Figure 4). In Tajima's *D* test, after the combination of four cpDNA regions of 373 Chinese *Zanthoxylum*, it was not significant at the *p* < 0.001 detection level, following the neutral model. To further investigate evolutionary relationships among 68 cpDNA haplotypes of Zanthoxylum accessions, we constructed a cpDNA haplotype network using NETWORK v.4.2.0.1 (Figure 5), the topology of which was perfectly consistent with the MCC tree.

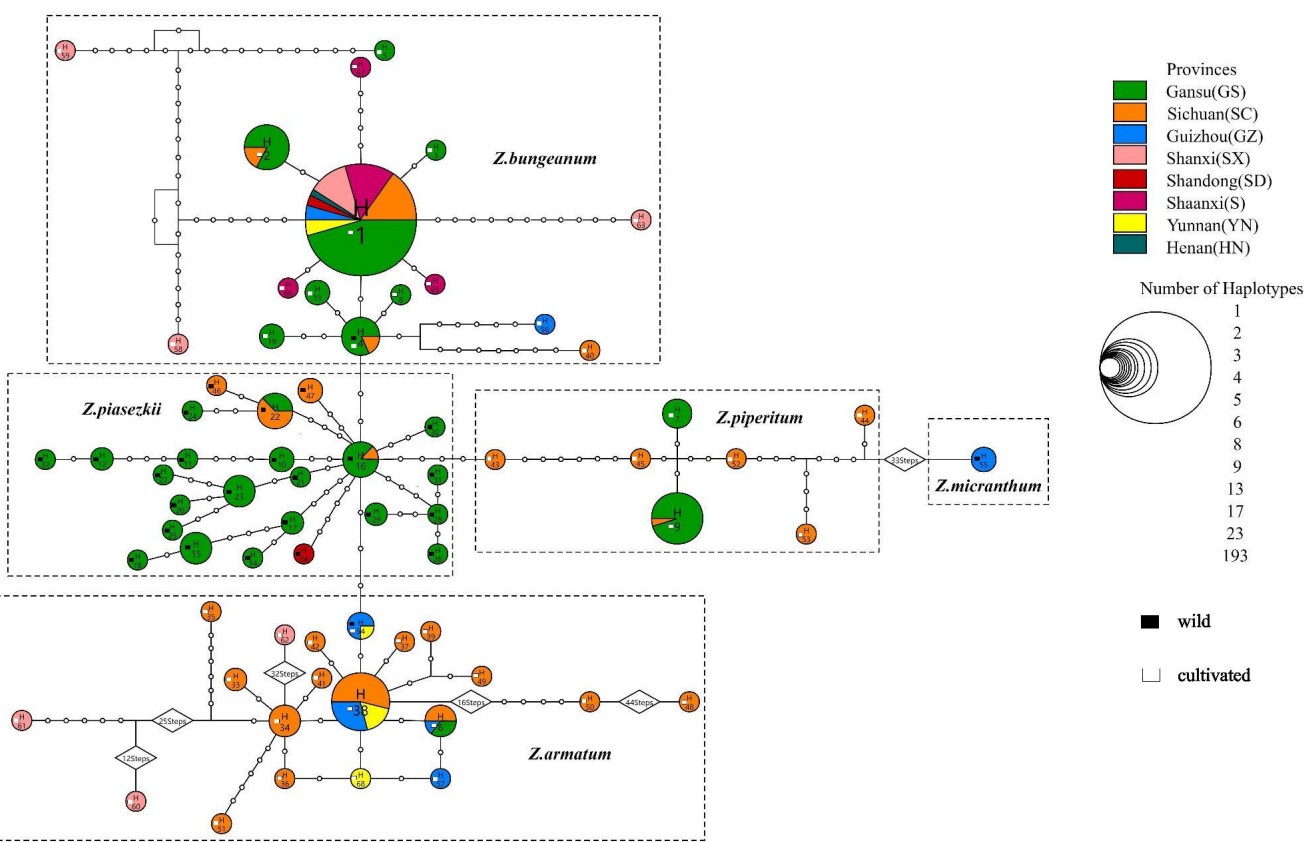

**Figure 5.** Median-joining network of 68 cpDNA haplotypes of *Zanthoxylum* accessions.

## 4. Discussion

### 4.1. Effect of Human Activity on the Genetic Structure of Zanthoxylum

Due to the fact that the Chinese *Zanthoxylum* has extremely high medicinal and edible value in China, farmers also began to cultivate a large number of high-quality *Zanthoxylum*. Thus, different *Zanthoxylum* species were widely cultivated in different places and developed a large number of hybrids during domestication. At present, 'Honghuajiao' (*Z. bungeanum*) and 'Qinghuajiao' (*Z. armatum*) are largely cultivated in China. 'Qinghuajiao' is mainly cultivated in the southwest of China. The neighbor-joining (NJ) tree showed that 'Dingtanhuajiao' and 'Tengjiao' were *Z. armatum*. We believe that the Chinese *Zanthoxylum* may originate from the domestication of wild *Zanthoxylum* species. In addition, we found some haplotypes of *Zanthoxylum* existed in different provenances. Even the distance between the two provenances is very far. This result may be caused by the transplantation of *Zanthoxylum* by farmers. Additionally, H1 was widely distributed in all provenances, and some *Zanthoxylum* species also had only one haplotype (such as H3, H21, H40, and H58).

This main reason was the effect of human activity. The cultivation history of *Z. bungeanum* was longer than *Z. armatum*. People have promoted the spread and exchange of genetic information in the process of transplantation and business and have thus affected the genetic structure of *Zanthoxylum*.

The nucleotide diversity and haplotype diversity should have decreased dramatically for cultivated populations compared with their wild ancestors in most crops [25]. We observed that *Z. piasezkii* had the highest haplotype and nucleotide diversity, while *Z. bungeanum* had the lowest (Table S2). It is possible that *Z. piasezkii* has a relatively short cultivation history and is mostly in the semi-wild state. Moreover, *Z. bungeanum* gradually became single during the long-term cultivation process, so the level of genetic diversity of *Z. bungeanum* was reduced. The genetic diversity of some cultivated *Zanthoxylum* species was high. This may be because *Zanthoxylum* had started apomixis. The reduction of the genetic diversity leads to weakened populations to adapt to their environment and evolutionary potential, which inevitably leads to poor population growth and makes the species critically endangered [26]. Therefore, it is urgent and important to strengthen the protection of wild *Zanthoxylum*.

*Zanthoxylum* is generally dioecious, but only a very few are monoecious and have apomixis characteristics (Fei 2021). However, it remains unclear whether *Zanthoxylum* can perform sexual reproduction. Our study was based on the collection of plant material carried out in situ germplasm. Although *Z. bungeanum* is widely distributed in China, most are cultivated *Zanthoxylum*. *Z. bungeanum* has few natural populations or is distributed in remote areas. This leads to the few wild germplasm that we can collect and makes sampling very difficult. Therefore, the results of this study may deviate from reality, and cannot fully reflect the genetic diversity of *Zanthoxylum* germplasm resources in China. Thus, we should continue to expand the sampling range in the future, and collect as many wild *Zanthoxylum* as possible.

### 4.2. The Origin of Cultivated Zanthoxylum Species

Moreover, the evolutionary network of haplotypes used in this study was an efficient approach that can be used to recover the directions of haplotypes transmissions at the local scale and spread at the larger scale, and particularly to identify the central haplotype that can be considered as the ancestral or founding haplotype [27]. Generally, ancestral or founding haplotypes are usually located at the center node of groups and are more frequently distributed. H1, H4, H16, H54, and H38 were the torso of the network figure, mainly distributed in Gansu and Sichuan. H16 was the center of the haplotype network, located in Zhouqu County, Gansu Province (Figure 5). Therefore, we inferred that 'Honghuajiao' (*Z. bungeanum*) may be distributed at the junction of Gansu and Sichuan provinces, greatly cultivated for the good shape and bright red color, and H16 was the center of origin of Chinese *Zanthoxylum*. This inference was the same as data that Gansu was the producing area of Chinese *Zanthoxylum* and concentrated in Tianshui, Longnan, and Wudu [28] [29]. H1 was derived from H16 and occurred in Gansu, Sichuan, Shaanxi, Shandong, Shanxi, Sichuan, Guizhou, and Yunnan. Given the geographic proximity, H1 most probably appeared first in Gansu and was introduced eastward by farmers. H38 originated in Gansu, Sichuan, and Yunnan and was derived directly from H16. In particular, Zhouqu belongs to Gannan Tibetan Autonomous Prefecture and is located in the southwest of Gansu Province. It connects between Tibet Plateau and Loess Plateau, Aba Prefecture in the south, and Longnan City and Linxia Prefecture in the east and north [30]. *Zanthoxylum* has a wide distribution range in China, and the continuous geographical distribution may increase the dispersal of seed and pollen flow between provenances. Thus, *Zanthoxylum* has a high level of genetic diversity.

We inferred that the cultivated *Zanthoxylum* species may originate in Zhouqu County, Gansu Province, and *Zanthoxylum* may expand from west to east in China. Historically, Gansu was a producing area of Chinese *Zanthoxylum* [31]. Meanwhile, the Qinghai–Tibet Plateau is located in the southwest of Gansu Province. It is known as the 'roof of the

world' and the 'third pole' of the earth and is the largest, highest, and youngest plateau in the world [32]. Additionally, the Qinghai–Tibet Plateau is also one of the global hot spots of biodiversity and an important 'gene bank' for plant resources in China [33]. The climate change of the ancient Mediterranean since the Tertiary and the uplift of the Qinghai–Tibet Plateau has caused a change in the distribution pattern of the ancient Mediterranean flora and even the global plants (Sun and Li 2003). It has been shown that the Chinese *Zanthoxylum* existed in the Late Cretaceous–Eocene plant flora in the early stages of the Qinghai–Tibet Plateau [34].

According to the analysis of BEAST, we found that the Chinese *Zanthoxylum* originated from at least the Late Eocene, and the most recent ancestor of the haplotypes of *Zanthoxylum* collected in this study was approximately the Middle Miocene. As early as 1985, Hongrong has already proposed that the Chinese *Zanthoxylum* was originally wild and cultivated in the Qinghai–Tibet Plateau [35]. Therefore, we inferred that the Chinese *Zanthoxylum* plants may originate from the Qinghai–Tibet Plateau.

## 5. Conclusions

*Zanthoxylum* genus has come to be a very valuable genus to the discovery and utilization of medicinal and agrochemical natural products. Sampling covered the main distribution regions of Chinese *Zanthoxylum* and was enough to represent the genetic resources of Chinese *Zanthoxylum*. According to the historical records, combined with the genetic structure, genetic diversity, and haplotype distribution characteristics, we inferred that cultivated *Zanthoxylum* species originated in Zhouqu County, Gansu Province, China, and Chinese *Zanthoxylum* originated from *Z. piasezkii*. Based on the ancestral divergence time of the cpDNA molecular level, we speculated that *Zanthoxylum* originated at least in the middle Tertiary Miocene, and the neutral test of cpDNA fragments indicated that the *Zanthoxylum* population did not undergo a significant expansion process. This result is consistent with the findings of Gregor, who expounded that *Zanthoxylum* first appeared in the Eocene–Cretaceous Period [36]. Thus, *Zanthoxylum* may be affected by the uplift, the Quaternary glacial period, and anthrogenic activities, expanding from west to east in China [36]. Therefore, we provided evidence for the origin of *Zanthoxylum* species in the Qinghai–Tibet Plateau of the southwest.

We should protect the natural wild *Zanthoxylum*, which was distributed in different regions, and the cultivated species of Chinese *Zanthoxylum* with excellent characteristics. We can take measures to combine local and ex situ protection, strengthen management, and reduce the interference of human activities. At the same time, we should also collect as many of the wild *Zanthoxylum* species as possible. These will be conducive to protecting the genetic diversity of Chinese *Zanthoxylum*.

However, the origin of Chinese *Zanthoxylum* was analyzed only based on the evidence of the existing population genetic diversity and the geographical distribution of germplasm resources combined with geohistorical data. This study can only be an initial guess. Chinese *Zanthoxylum* remains to be studied deeply to provide new insights into the phylogenetic relationships between wild and cultivated *Zanthoxylum* species in China.

**Supplementary Materials:** The following supporting information can be downloaded at: https://www.mdpi.com/article/10.3390/horticulturae8050440/s1, Table S1: List of Zanthoxylum accessions used in this study; Table S2: Haplotype composition, genetic diversity (Hd), nucleotide diversity (Pi), Tajima's D, and Fu's Fs of 373 accessions of Zanthoxylum. Figure 3 Neighbor-joining (NJ) tree of 373 wild and cultivated Zanthoxylum accessions based on the cpDNA sequences. Figure 4 The maximum clade credibility tree (MCC) was generated based on the dataset of 68 cpDNA haplotypes of 373 Zanthoxylum accessions. Figure 5 Median - joining network of 68 cpDNA haplotypes of Zanthoxylum.

**Author Contributions:** Conceptualization, X.C. and L.T.; methodology, S.F.; software, J.T.; validation, X.C. and L.T.; formal analysis, S.F.; investigation, G.W. and X.G.;resources, A.W.; data curation, X.C.; writing—original draft preparation, X.C.; writing—review and editing, X.C.; visualization, X.C.; supervision, A.W. and S.F.; project administration, A.W.; funding acquisition, A.W. All authors have read and agreed to the published version of the manuscript.

**Funding:** This research was funded by [Technology Innovation Leading Program of Shaanxi] grant number [2020QFY07-01] and Biosafety and Genetic Resource Management Project grant number [KJZXSA202025].

**Institutional Review Board Statement:** We choose to exclude this statement and the study did not require ethical approval.

**Informed Consent Statement:** The study did not involve humans.

**Data Availability Statement:** We choose to exclude this statement and the study did not report any data.

**Acknowledgments:** This work was supported by the Technology Innovation Leading Program of Shaanxi (2020QFY07-01), the Biosafety and Genetic Resource Management Project (KJZXSA202025), Project Supported by the Department of Science and Technology of Guizhou Province ([2021]222), and the Guizhou Kehe Platform Talent ([2019]5643). We would like to thank Yinming Wu and his team at the Sichuan Academy of Botanical Engineering, Zongxing Wu, and Hui Xu of the Sichuan Academy of Forestry for help with sampling.

**Conflicts of Interest:** The authors declare no conflict of interest.

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
