# Peer review of "Extensive Sampling Provides New Insights into Phylogenetic Relationships between Wild and Domesticated Zanthoxylum Species in China"

_horticulturae, doi:10.3390/horticulturae8050440_

Round 1

Reviewer 1 Report

The authors Xue Chen et al in their paper entitled “Extensive sampling provides new insights into phylogenetic relationships between wild and domesticated Zanthoxylum species in China” have shown in their manuscript the phylogenetic relationship and the origin of wild and cultivated Zanthoxylum species in China. The authors use cpDNA to deduce the genetic diversity, relatedness and the geographical origin of this species. While this study attempts at getting down to the origin of the species it additionally also promotes identification and breeding programs and techniques that can be adopted for an enhanced economical advantage. The manuscript is well structured however, I have the following minor concerns.

Minor Concerns

  1. The authors can improve on the overall writing and sentence structure and formation. In particular the authors can look into the line 50 to 65 and revamp their sentence formations. I would also suggest the authors to revisit the manuscript many time and have it checked by a native speaker.
  2. In Line 117 the authors describe the PCR protocol and use the word 1ul. This term can be misleading and does not direct the quantity of the template used. I suggest the authors mention the concentration of genomic DNA used instead.
  3. Additionally, while the authors attempt at describing the locations from where their samples were taken and also strongly supported with geographical locations and ArcGIS I would be very curious also if the authors could also present any phylogenetic changes that they observed with respect to the different samples taken from different geographical locations across China. The authors did mention about the good shape and bright red colour. Probably a pictorial representation would help the readers with connecting to the plant diversity and a table highlighting the economical benefits and phenotypes could also be highly informational.

Overall, I feel the manuscript is well written and the authors could help in addressing the minor concerns raised above. This would help the overall structure and readership as well as highlight the relevance of the research carried out.

Author Response

  Thank you for the reviewers' comments concerning our manuscript. Those  comments are all valuable and very helpful for revising and improving our paper, as well as the important guiding significance to our researches. We have studied comments carefully and have made correction which we hope meet with approval. Revised portion are marked in red in the paper. The main corrections and the responds to the reviewers' comments are as flowing:

Reviewer 2 Report

Notes on Introduction: The research sought to characterize germplasm of a species reported as important. The definition of an important species was basically related to the properties of secondary compounds present in the plant. It is not quantitatively evident the importance of the species in socioeconomic and industrial terms in an attempt to better justify the research objectives. Additionally, obsolete citations in the text are observed (Huang 1997, Wang et al. 2002, Guo et al. 2011, Badenes and Parfitt 1995; Xu et al. 2001). It is suggested that the text of the introduction be reformulated with more recent citations. A critical point identified was the lack of information about the reproductive system of the species. Need to write that germplasm is of the in situ conservation type. These important observations were not identified as the manuscript deals with germplasm/characterization.

Material and methods: Lack of further details of the methodology. The writing became very objective, preventing reproduction by other researchers. It needs to detail the molecular procedures and the way to obtain the molecular values.

Results and Discussion: It is very well written. Well-designed and planned figures to facilitate the understanding of the results. One fact stands out. The results and the discussion were based on the collection of plant material carried out in in situ germplasm. It is not evident in the text an alert about issues related to the reproductive system of the species. What is the consequence if you collect from other plants even though they are from the same region? it is a mixture of pure lines (for autogamous) or a population (for allogamous). Information from the reproductive system is very important in all parts of the research to make the results clearer. The type of reproductive system, morphological or biochemical adaptations may change the result if further research is carried out. This needs to be explained in the text to be able to limit writing in this information. The conclusion could be more objective.

Author Response

(The authors gave the same response as above.)

Reviewer 3 Report

Extensive sampling provides new insights into phylogenetic relationships between wild and domesticated Zanthoxylum species in China

Manuscript ID: horticulturae-1672975

Submitted to Horticulturae

The manuscript " Extensive sampling provides new insights into phylogenetic relationships between wild and domesticated Zanthoxylum species in China” presents a molecular phylogenetic analysis of wild and cultivated Zanthoxylum species and evolutionary relationships between Zanthoxylum species. An attempt was also made to determine the routes of spread of Zanthoxylum species in China. The authors used three modern methods in their study: maximum likelihood (ML), neighborjoining (NJ) and Bayesian Inference (BI).

The structure of the manuscript, as well as presentation of results, and the style of the paper are appropriate for a scientific journal. However, the authors have not avoided errors and shortcomings that need to be corrected.

The Abstract is adequate to the content.

The Introduction describes the research issues and the aim of the studies, however, authors did not refer to the manuscript by Feng et al. 2016 Tree Genetics & Genomes 12: 45, describing the issue of phylogenetic relationship between Zanthoxylum species in China.

L43 - …………. has been used to development of new antitumor drugs. (Francesco et al. 2010)

I suggest changing to ………………has been used to develop of new antitumor drugs……………..

L44 - The citation Guo et al.  2011 is given, while in the reference list is Guo et al. 2010

L46 - Pan et al. 2017 is missing in the reference list

L46 - ……the cultivated Zanthoxylum species is suitable for planting……

I suggest changing to …………..Zanthoxylum species are suitable for planting……..

L50 - Z. bungeanum is native to southwestern China and has been cultivated and cultivated for more than………

L55-58 – For example, the pericarps color of Z. bungeanum and Z. armatum is bright red or green, and therefore, these cultivars are commonly known as ………

From the sentence above, it appears that the Zanthoxylum bungeanum and Z. armatum listed, are cultivars, whereas these are species. Please, make it clear.

L59-61 - Furthermore, some cultivars or varieties that share similar morphological characteristics often have the same names, but virtually belong to different species.

How do you understand the difference between a cultivar and variety? A single term for cultivated forms should be chosen.

L72-74 For example, Wang et al. (2021) analyzed the genetic diversity of sweet potato germplasm based on three cpDNA spacer sequences, trnL-trnF, trnH-psbA, and trnT-trnL.

Wang manuscripts from 2021 should be marked with an additional sign e.g. Wang et al. 2021a or Wang et al. 2021b

The plant material is not sufficiently described. The authors report that it contained 373 accessions and provide a reference to Tables S1 and S2 but these tables were not included.

The authors state that the name of each cultivar (accession) was designated by local farmers. It is not a reliable way to determine the systematic affiliation of plants. In my opinion, taxonomic keys should be used to identify unknown accessions. Furthermore, the authors mentioned that classification of species of Zanthoxylum based solely on morphological characteristics, is difficult.

It is necessary to complete the Materials and Methods section.

L 115 - What was the studied material? Were the leaves fresh or dried?

L117 and 122 Please, specify producers of MasterMix and agarose gel

The results described are complete, but they refer to figures which are unreadable due to the font being too small. It refers to Figure 2, 3, 4.

The figure legends need a revision, so that they can be understood without recourse to the body of the manuscript. First of all, the authors should give the name of the species or genus that was studied. In my opinion they could also expand the abbreviations MCC, Ma in the Figure description.

I recommend completing the geological time scale on the Figure 3 with the names of the epochs

L166 - Insert a space between: 3 Indels (Insertions and Deletions)

L177 - For Z. armatum, the haplotype H1 was the most (193/373), including wild and cultivated Zanthoxylum accessions from all sampled…………….

Suggested re-wording: For Z. armatum, the haplotype H1 was the most numerous (193/373), including wild and cultivated…………………

L202 and 203 Insert a space between: Group II was…………,    Group III was Z. piasezkii…………………

L 215 Figure 2. Neighbor-joining (NJ) tree of 373 species wild and cultivated Zanthoxylum species based on the cpDNA sequences.

373 accessions were studied, not species.

L222 For Z. piasezkii, mainly divided into four branches

Suggested re-wording: Z. piasezkii was mainly divided ………

L228 and L230 ……………………..the first part was again divided into three subclades, includes ten haplotypes derived from all sampling provinces (H67, H1, H3, H63; H5, H59, H58; H4, H56, H2), and the second portion, includes ten haplotypes derived from Gansu

Suggested re-wording: includes change into including

L247 Insert please, what does the blue lines on the figure mean? Enlarge the font in the figure.

L249 - Figure 3. MCC tree from the BEAST analysis based on the dataset of four cpDNA markers of 68 haplotypes, Taddalia asiatica, Citrus limon, Citrus sinensis were used as outgroups.

68 haplotypes of what? Zanthoxylum ??? Please specify it.

L253 - Figure 4. Median - joining network of cpDNA 68 haplotypes.

68 haplotypes of what? Zanthoxylum ??? Please, specify it.

The discussion needs thorough revision. The dominant part of this chapter was devoted to summarizing the results. In my opinion, there is no sufficient interpretation of the results especially with regard to literature.

L256-262 Repetition of the description of the results. Add an interpretation

L259-260 An incomprehensible sentence. Mental shortcut. Please, correct it.

L262 - Owing to the Chinese Zanthoxylum has extremely high medicinal and edible value in China, farmers also began to cultivate a large number of high quality Zanthoxylum.

Suggested re-wording: Due to the fact that the Chinese Zanthoxylum has extremely high…………………………………..

The subsection 4.1 “Effect of human activity on the genetic structure of Zanthoxylum” was discussed a little briefly and needs to be expanded or rewritten.

Please, indicate clearly the human influence on the genetic structure of Zanthoxylum.

L 276-277 – Cite literature.

L280-281 We observed that Gansu had higher haplotype and nucleotide diversity.

Please, explain it. Give a reason why there was more diversity in the province Gansu.

L290 This inference was the same as data that Gansu was the producing area of Chinese Zanthoxylum and concentrated in Tianshui, Longnan, and Wudu (Crandall and Templeton 1993).

Is there good reference?

L294-297 Especially, Zhouqu belongs to Gannan Tibetan Autonomous Prefecture located in the southwest of Gansu Province, connects between Gansu, Tibet Plateau, and Loess Plateau. While Aba Prefecture in the south, and Longnan City and Linxia Prefecture in the east and north (Sun 2002).

What are the connections between the presented fragment and the obtained results. Please, specify it.

L306-308 We could obtain a rough estimate that the closest ancestor of the haplotype of Zanthoxylum plants was approximately 20.00 Ma, probably the Middle Miocene based on BEAST analysis.

To support it, it would be good to include the names of the epochs on the Figure 3

Improvement of the English language is necessary.

Author Response

(The authors gave the same response as above.)

Round 2

Reviewer 1 Report

I thank the authors of the manuscript for incorporating the corrections suggested. I believe this will enhance the overall readership  and will also help to deliver a clearer message. The current version of the manuscript is acceptable.

Author Response

We are very grateful to the reviewers for accepting the current version of

the manuscript.

Reviewer 2 Report

The authors improved the manuscript. The authors chose not to accept the suggestion to add more details regarding the reproductive system. This can limit information for readers.

Author Response

We thank the reviewer for these comments which will definitely help us to improve our manuscript. This is a constructive suggestion. Although we have found little literature about the reproductive system of Zanthoxylum, we have tried to add these contents in the Discussion sector of this revision, which we hope meet with approval.

Reviewer 3 Report

Extensive sampling provides new insights into phylogenetic relationships between wild and domesticated Zanthoxylum species in China

Manuscript ID: horticulturae-1672975

Submitted to Horticulturae

The authors of the manuscript " Extensive sampling provides new insights into phylogenetic relationships between wild and domesticated Zanthoxylum species in China” responded to comments and suggestions. They made changes to the text and presented a revised version of the manuscript. However, the text still requires minor revision. The relevant comments are included on the PDF file of the manuscript.

Introduction: “However, there is no reasonable explanation for the origin of these cultivars and their evolutionary relationship (Feng 2016)”

In my opinion, the evolutionary relationship is contained in the Feng et al. 2016 manuscript.

The discussion needs improvement in two places. I recommend that the authors rewrite the text.

Please write what is the consequence the "Qinghai-Tibet Plateau is one of the global hot spots of biodiversity and .............." Combine it with your own results.

The figure legends need a revision, so that they can be understood without recourse to the body of the manuscript. First of all, the authors should give the name of the species or genus that was studied. In my opinion they could also expand the abbreviations MCC, Ma in the Figure description.

The authors described in cover letter that they completed the geological time scale on the Figure 3 (current Figure 4) with the names of the Epochs. I cannot find these names on the figure, at least in the version of the file I received.

Author Response

1. Responseto comment: (Introduction: “However, there is no reasonable explanation for the origin of these cultivars and their evolutionary relationship (Feng 2016)”)

Response: We are very sorry for this kind of mistakes to eliminate such kind ofproblems. We have redescribed the issues ofphylogenetic relationships between Zanthoxylum in China with reference to Feng's articles. 

2. Responseto comment: (Please write what is the consequence the "Qinghai-Tibet Plateau is one of the global hot spots of biodiversity and .............." Combine it with your own )

Response: We have re-written this part according to the Reviewer’s suggestion.

3. Response to comment: (The figure legends need a revision.)

Response: We have made correction according to the Reviewer’ suggestions.

Fig. 4 The maximum clade credibility tree (MCC) was generated based on the dataset of 68 cpDNA haplotypes of 373 Zanthoxylum accessions. Taddalia asiaticaCitrus limonCitrus sinensis were used as outgroups. Posterior probabilities based on maximum likelihood (ML) analysis are labeled above nodes. Blue bars represent the 95 % highest posterior density (HPD) credibility intervals for node ages (in Myr ago,  Ma).

4. Response to comment: (The authors described in cover letter that they completed the geological time scale on the Figure 3 (current Figure 4) with the names of the Epochs. I cannot find these names on the figure, at least in the version of the file I received.)

Response: We are very sorry for the last figure upload error and have re-uploaded the correct figure 4. And we have completed the geological time scale on the Figure 4 with the names of the Epochs at the bottom of Fig.

Other changes:

  1. Thestatements of “The sequences of all the DNA sequences … …”were corrected as “The sequences of all the DNA fragment … …”
  2. Thestatements  of “Moreover,  the  evolutionary  network  of haplotypes  was  an efficient … …” were corrected as “Moreover, the evolutionary network of haplotypes used in this study was an efficient … …”

We  tried  our  best  to  improve  the  manuscript  and  made  some  changes  in  the manuscript. These changes will not influence the content and framework of thepaper. And here we listed the changes and marked in revised paper.

We appreciate for Editors and Reviwers’ warm work earnestly, and hope that the correction  will  meet  with  approval.  Once  again,  thank  you  very  much  for  your comments and suggestions.
